# Self-Labeling the Job Shop Scheduling Problem

**Andrea Corsini    Angelo Porrello    Simone Calderara    Mauro Dell'Amico**

University of Modena and Reggio Emilia, Italy
`{name.surname}@unimore.it`

## Abstract

This work proposes a self-supervised training strategy designed for combinatorial problems. An obstacle in applying supervised paradigms to such problems is the need for costly target solutions often produced with exact solvers. Inspired by semi- and self-supervised learning, we show that generative models can be trained by sampling multiple solutions and using the best one according to the problem objective as a pseudo-label. In this way, we iteratively improve the model generation capability by relying only on its self-supervision, eliminating the need for optimality information. We validate this *Self-Labeling Improvement Method (SLIM)* on the Job Shop Scheduling (JSP), a complex combinatorial problem that is receiving much attention from the neural combinatorial community. We propose a generative model based on the well-known Pointer Network and train it with SLIM. Experiments on popular benchmarks demonstrate the potential of this approach as the resulting models outperform constructive heuristics and state-of-the-art learning proposals for the JSP. Lastly, we prove the robustness of SLIM to various parameters and its generality by applying it to the Traveling Salesman Problem.

## 1   Introduction

The Job Shop Problem (JSP) is a classic optimization problem with many practical applications in industry and academia [56]. At its core, the JSP entails scheduling a set of jobs across multiple machines, where each job has to be processed on the machines following a specific order. The goal of this problem is generally to minimize the *makespan*: the time required to complete all jobs [40].

Over the years, various approaches have been developed to tackle the JSP. A common one is adopting exact methods such as Mixed Integer Programming solvers (MIP). However, these methods often struggle to provide optimal or high-quality solutions on medium and large instances in a reasonable timeframe [29]. As a remedy, metaheuristics have been extensively investigated [36, 13, 24] and constitute an alternative to exact methods. Although state-of-the-art metaheuristics like [37] can rapidly provide quality solutions, typically within minutes, they are rather complex to implement and their results can be difficult to reproduce [6]. Therefore, due to their lower complexity, Priority Dispatching Rules (PDR) [40] are frequently preferred in practical applications with tighter time constraints. The main issue of PDRs is their tendency to perform well in some cases and poorly in others, primarily due to their rigid, hand-crafted prioritization schema based on hardcoded rules [21].

Recent works have increasingly investigated Machine Learning (ML) to enhance or replace some of these approaches, primarily focusing on PDRs. ML-based approaches, specifically Reinforcement Learning (RL) ones, proved to deliver higher-quality solutions than classic PDRs at the cost of a small increase in execution time [33]. Despite the potential of RL [54, 25], training RL agents is a complex task [45, 34], is sensitive to hyper-parameters [43], and has reproducibility issues [22].

Contrary, supervised learning is less affected by these issues but heavily relies on expensive annotations [32]. This is particularly problematic in combinatorial problems, where annotations in the

38th Conference on Neural Information Processing Systems (NeurIPS 2024).

form of (near-) optimal information are generally produced with expensive exact solvers. To contrast labeling cost and improve generalization, *semi-supervised* [48] and *self-supervised* [32] are gaining popularity in many fields due to their ability to learn from unlabeled data. Despite this premise, little to no application of these paradigms can be found in the JSP and the combinatorial literature [5, 9].

Motivated by these observations, we introduce a novel self-supervised training strategy that is simpler than most RL algorithms, does not require the formulation of the Markov Decision Process [45], and relies only on model-generated information, thus removing the need for expensive annotations. Our proposal is based on two weak assumptions: i) we suppose to be able to generate *multiple solutions* for a problem, a common characteristic of generative models [38]; and ii) we suppose it is possible to *discriminate solutions based on the problem objective*. When these assumptions are met, we train a model by generating multiple solutions and using the best one according to the problem objective as a pseudo-label [31]. This procedure draws on the concept of pseudo-labeling from semi-supervised learning, but it does not require any external and expansive annotation as in self-supervised learning. Hence, we refer to it as a *SLIM: Self-Labeling Improvement Method.*

We prove the effectiveness of SLIM primarily within the context of Job Shop, a challenging scheduling problem with many baseline algorithms [1, 36] and established benchmarks [44, 46, 16]. As recognized in other works [54, 39, 25], focusing on the JSP is crucial because its study helps address related variants, such as Dynamic JSP [51] and Flow Shop [42], while also establishing a concrete base for tackling more complicated scheduling problems like Flexible JSP [53].

Similar to PDRs, we cast the generation of solutions as a sequence of decisions, where at each decision one job operation is scheduled in the solution under construction. This is achieved with a generative model inspired by the *Pointer Network* [50], a well-known architecture for dealing with sequences of decisions in combinatorial problems [5]. We train our model on random instances of different sizes by generating multiple parallel solutions and using the one with the *minimum makespan* to update the model. This training strategy produces models outperforming state-of-the-art learning proposals and other conventional JSP algorithms on popular benchmark sets. Furthermore, our methodology demonstrates robustness and effectiveness across various training regimes, including a brief analysis on the Traveling Salesman Problem to emphasize the broader applicability of SLIM beyond scheduling. In summary, the contributions of this work are:

- Our key contribution is the introduction of SLIM, a novel self-labeling improvement method for training generative models. Thanks to its minimal assumptions, SLIM can be easily applied as-is to other combinatorial problems.

- Additionally, we present a generative encoder-decoder architecture capable of generating high-quality solutions for JSP instances in a matter of seconds.

The remainder of this work is organized as follows: Sec. 2 reviews related ML works; Sec. 3 formalizes the JSP; Sec. 4 introduces our generative model and SLIM; Sec. 5 compares our approach with others; Sec. 6 studies additional aspects of our proposal; and Secs. 7 and 8 close with general considerations, some limitations, and potential future directions.

## 2 Related Works

ML approaches have been recently investigated to address issues of traditional JSP methodologies.

Motivated by the success of **supervised learning**, various studies have employed exact methods to generate optimality information for synthetic JSP instances, enabling the learning of valuable scheduling patterns. For example, [26] used imitation learning [5] to derive superior dispatching rules from optimal solutions, highlighting the limitations of learning solely from optimal solutions. [28] proposed a joint utilization of optimal solutions and Lagrangian terms during training to better capture JSP constraints. Whereas, in [14], the authors trained a Recurrent Neural Network (RNN) to predict the quality of machine permutations – generated during training with an MIP – for guiding a metaheuristic. Despite their quality, these proposals are limited by their dependence on costly optimality information.

To avoid the need for optimality, other works have turned to **reinforcement learning**. The advantage of RL lies in its independence from costly optimal information, requiring only the effective formulation of the Markov Decision Process [45]. Several works focused on creating better neural PDRs, showcasing the effectiveness of policy-based methods [54, 39, 25, 11]. All these methods adopted a

Proximal Policy Optimization algorithm [43] and proposed different architectures or training variations. For instance, [54, 39] adopted Graph Neural Networks (GNN), [11] proposed a rather complex Transformer [49], while [25] used RNNs and curriculum learning [58]. Differently, [20] presented a Double Deep Q-Network approach, proving the applicability of value-based methods.

Other works employed RL to enhance other algorithmic approaches. In [12], an iterative RL-based approach was proposed to rewrite regions of JSP solutions selected by a learned policy. Whereas [17] utilized a Deep Q-learning approach to control three points of interventions within metaheuristics, and [55] recently presented an RL-guided improvement heuristic. Differently, [47] presented a hybrid imitation learning and policy gradient approach coupled with Constraint Programming (CP) for outperforming PDRs and a CP solver.

## 3  Job Shop Scheduling

In the *Job Shop Problem*, we are given a set of jobs $J = \{J_1, \ldots, J_n\}$, a set of machines $M = \{M_1, \ldots, M_m\}$, and a set of operations $O = \{1, \ldots, o\}$. Each job $j \in J$ comprises a sequence of $m_j$ consecutive operations $O_j = (l_j, \ldots, l_j + m_j - 1) \subset O$ indicating the order in which the job must be performed on machines, where $l_j = 1 + \sum_{i=1}^{j-1} m_i$ is the index of its first operation (e.g., $l_1 = 1$ for $J_1$ and $l_2 = 3$ for $J_2$ in Fig. 1). An operation $i \in O$ has to be performed on machine $\mu_i \in M$ for an uninterrupted amount of time $\tau_i \in \mathbb{R}_{\geq 0}$. Additionally, machines can handle one operation at a time. The objective of the JSP is to provide an order to operations on each machine, such that the precedences within operations are respected and the time required to complete all jobs (**makespan**) is minimized. Formally, we indicate with $\pi$ a JSP solution comprising a permutation of operations for each machine and use $C_i(\pi)$ to identify the completion time of operation $i \in O$ in $\pi$.

A JSP instance can be represented with a **disjunctive graph** $G = (V, A, E)$ [3], where: $V$ contains one vertex for each operation $i \in O$; $A$ is the set of directed arcs connecting consecutive operations of jobs reflecting the order in which operations must be performed; $E$ is the set of *disjunctive* (undirected) edges connecting operations to perform on the same machines. When the JSP is represented as a disjunctive graph $G$, finding a solution means providing a direction to all the edges in $E$, such that the resulting graph is directed and acyclic (all precedences are respected). As in related works [54, 39], we augment the disjunctive graph by associating to each vertex a set of 15 features $x_i \in \mathbb{R}^{15}$ describing information about operation $i \in O$. For lack of space, we present these features in App. A.

### 3.1  Constructing Feasible Job Shop Solutions

Feasible JSP solutions can be constructed step-by-step as a *sequence of decisions*, a common approach adopted by PDRs [40] and RL algorithms [54, 25]. The construction of solutions can be visualized using a branch-decision tree, shown at the bottom of Fig. 1. Each path from the root node ($R$) to a leaf node presents a particular sequence of $o = |O|$ decisions and leads to a valid JSP solution, such as the one highlighted in gray. At every node in the tree, one uncompleted job needs to be selected, and its ready operation is scheduled in the solution being constructed. Due to the precedence constraints and the partial solution $\pi_{<t}$ constructed up to decision $t$, there is at most one operation $o(t, j) \in O_j$ that is ready to be scheduled for any job $j \in J$. Thus, by selecting a job $j$, we uniquely identify an operation $o(t, j)$ that will be scheduled by appending it to the partial permutation of its machine. Notice that once a job is completed, it cannot be selected again, and such a situation is identified with a cross in Fig. 1.

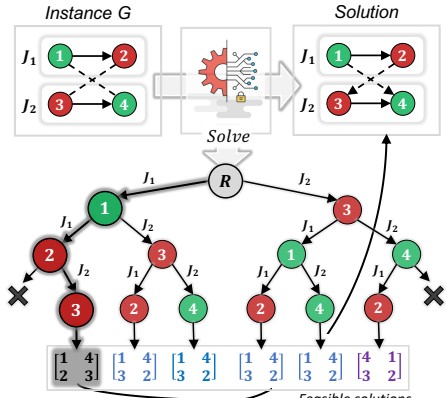

Figure 1: The sequences of decisions for constructing solutions in a JSP instance with two jobs ($J_1$ and $J_2$) and two machines (identified in green and red). Best viewed in colors.

We emphasize that diverse paths can lead to the same solution, indicating certain solutions are more frequent than others. In some instances, near-optimal solutions may lie at the end of unlikely and hardly discernible paths, making their construction more challenging. This could explain why PDRs,

Table 1: The features of a context vector $c_j \in \mathbb{R}^{11}$ that describes the status of a job $j$ within a partial solution $\pi_{<t}$. Recall that $o(t, j)$ is the ready operation of job $j$ at step $t$, $\mu_{o(t, j)}$ its machine, and $o(t, j) - 1$ its predecessor.

| ID | Description | Rationale |
|---|---|---|
| 1 | $C_{o(t, j)-1}(\pi_{<t})$ minus the completion time of machine $\mu_{o(t, j)}$. | The idle time of job $j$ if scheduled on its next machine $\mu_{o(t, j)}$ at time $t$. Negative if the job could have started earlier. |
| 2 | $C_{o(t, j)-1}(\pi_{<t})$ divided by the makespan of $\pi_{<t}$. | How close job $j$ is to the makespan of the partial solution $\pi_{<t}$. |
| 3 | $C_{o(t, j)-1}(\pi_{<t})$ minus the average completion time of all jobs. | How early/late job $j$ completes w.r.t. the average among all jobs in $\pi_{<t}$. |
| 4 - 6 | The difference between $C_{o(t, j)-1}(\pi_{<t})$ and the $1^{st}$, $2^{nd}$, and $3^{rd}$ quartile computed among the completion time of all jobs. | Describe the relative completion of $j$ w.r.t. other jobs. These features complement that with ID = 3. |
| 7 | The completion time of machine $\mu_{o(t, j)}$ divided by the makespan of the partial solution $\pi_{<t}$. | How close the completion time of machine $\mu_{o(t, j)}$ is to the makespan of the partial solution $\pi_{<t}$. |
| 8 | The completion time of machine $\mu_{o(t, j)}$ minus the average completion of all machines in $\pi_{<t}$. | How early/late machine $\mu_{o(t, j)}$ completes w.r.t. the average among machines in $\pi_{<t}$. |
| 9 - 11 | The difference between the completion of $\mu_{o(t, j)}$ and the $1^{st}$, $2^{nd}$, and $3^{rd}$ quartile among the completion time of all machines. | Describe the relative completion time of machine $\mu_{o(t, j)}$ w.r.t. all the other machines. These features complement that with ID = 8. |

which rely on rigid, predefined rules, sometimes struggle to generate quality solutions, whereas neural algorithms – able to leverage instance- and solution-wide features – may perform better.

## 4 Methodology: Generative Model & Self-Labeling

As described in Sec. 3.1, we tackle the JSP as a sequence of decisions. Thus, we propose a generative model inspired by the Pointer Network [50] (a well-known encoder-decoder architecture to generate sequences of decisions) whose goal is to select the right job at each decision step $t$. Formally, our model learns a function $f_\theta(\cdot)$, parametrized by $\theta$, that estimates the probability $p_\theta(\pi|G)$ of a solution $\pi$ being of high-quality, expressed as a product of probabilities:

$$p_\theta(\pi|G) = \prod_{t=1}^{|O|} f_\theta(y_t \mid \pi_{<t}, G), \tag{1}$$

where $f_\theta(y_t \mid \pi_{<t}, G)$ gives the probability of selecting job $y_t \in J$ for creating $\pi$, conditioned on the instance $G$ and the partial solution $\pi_{<t}$ constructed up to step $t$. By accurately learning $f_\theta(\cdot)$, the model can construct high-quality solutions autoregressively. The following sections present the proposed encoder-decoder architecture and the self-labeling strategy to train this generative model.

### 4.1 The Generative Encoder-Decoder Architecture

**Overview.** Our model processes JSP instances represented as a disjunctive graph $G$ and produces single deterministic or multiple randomized solutions depending on the adopted construction strategy. The *encoder* operates at the instance level, creating embedded representations for all the operations in $O$ with a single forward computation of $G$ (hence, no autoregression). Whereas the *decoder* operates at the solution level (job and machine level), using the embeddings of ready operations and the partial solution $\pi_{<t}$ to produce a probability of selecting each job at step $t$. Recall that after selecting a job $j$, its ready operation $o(t, j)$ is scheduled in $\pi_{<t}$ to produce the partial solution for step $t + 1$.

**Encoder.** It captures instance-wide relationships into the embeddings of operations, providing the decoder with a high-level view of the instance characteristics. The encoder can be embodied by any architecture, like a Feedforward Neural Network (FNN), that transforms raw operation features $x_i \in \mathbb{R}^{15}$ (see App. A for the features) into embeddings $e_i \in \mathbb{R}^h$. As in related works [54, 39], we also encode the relationships among operations present in the disjunctive graph. We thus equip the encoder with Graph Neural Networks [52], allowing the embeddings to incorporate topological information of $G$. In our encoder, we stack two Graph Attention Network (GAT) layers [8] as follows:

$$e_i = [x_i \,||\, \sigma(\text{GAT}_2(\,[x_i \,||\, \sigma(\text{GAT}_1(x_i, \, G))], \, G)], \tag{2}$$

where $\sigma$ is the ReLU non-linearity and $||$ stands for the concatenation operation.

**Decoder.** It produces at any step $t$ the probability of selecting each job from the embeddings $e_i$ and solution-related features. The encoder is logically divided into two distinct components:

- *Memory Network*: generates a state $s_j \in \mathbb{R}^d$ for each job $j \in J$ from the partial solution $\pi_{<t}$. This is achieved by first extracting from $\pi_{<t}$ a context vector $c_j$ (similarly to [27]), which contains

eleven hand-crafted features providing useful cues about the status of job $j$. We refer to Tab. 1 for the definition and meaning of these features. Then, these vectors are fed into a Multi-Head Attention layer (MHA) [49] followed by a non-linear projection to produce jobs' states:

$$s_j = \text{ReLU}([c_j\,W_1 + \underset{b \in J}{\text{MHA}}(c_b\,W_1)]\,W_2), \tag{3}$$

where $W_1$ and $W_2$ are projection matrices. Note that we use the MHA to consider the context of all jobs when producing the state for a specific one, similarly to [11].

- *Classifier Network*: outputs the probability $p_j$ of selecting a job $j$ by combining the embedding $e_{o(t,\,j)}$ of its ready operation and the state $s_j$ produced by the memory network. To achieve this, we first concatenate the embeddings $e_{o(t,\,j)}$ with the states $s_j$ and apply an FNN:

$$z_j = \text{FNN}([e_{o(t,\,j)}\,\|\,s_j]). \tag{4}$$

Then, these scores $z_j \in \mathbb{R}$ are transformed into softmax probabilities: $p_j = e^{z_j}\,/\,\sum_{b \in J} e^{z_b}$. The final decision on which job to select at $t$ is made using a sampling strategy, as explained next.

**Sampling solutions.** To generate solutions, we employ a probabilistic approach for deciding the job selected at any step. Specifically, we randomly sample a job $j$ with a probability $p_j$, produced at step $t$ by our decoder. We also prevent the selection of completed jobs by setting their scores $z_j = -\infty$ to force $p_j = 0$ before sampling. Note how this probabilistic selection is *autoregressive*, meaning that sampling a job depends on $p_j$ which is a function of $e_{o(t,\,j)}$ and $s_j$ resulting from earlier decisions (see Eqs. 3 and 4). Various strategies exist for sampling from autoregressive models, including top-k [23], nucleus [23], and random sampling [4]. Preliminary experiments revealed that these strategies perform similarly, with average optimality gap variations within $0.2\%$ on benchmark instances. However, top-k and nucleus sampling introduce brittle hyperparameters that can hinder training convergence if not managed carefully. Therefore, we adopt the simplest strategy for training and testing, which is *random sampling*: sampling a job $j$ with probability $p_j$ at random. Note however that the sampling strategy remains a flexible design choice, not inherently tied to our methodology.

## 4.2 SLIM: Self-Labeling Improvement Method

We propose a self-supervised training strategy that uses the model's output as a teaching signal, eliminating the need for optimality information or the formulation of Markov Decision Processes. Our strategy exploits two aspects: the capacity of generative models to construct multiple (parallel) solutions and the ability to discriminate solutions of combinatorial problems based on their objective values, such as the makespan in the JSP. With these two ingredients, we design a procedure that at each iteration generates multiple solutions and uses the best one as a *pseudo-label* [31].

More in detail, for each training instance $G$, we randomly sample a set of $\beta$ solutions from the generative model $f_\theta(\cdot)$. We do that by keeping $\beta$ partial solutions in parallel and independently sample for each one the next job from the probabilities generated by the model. Once the solutions have been completely generated, we take the one with the minimum makespan $\bar{\pi}$ and use it to compute the Self-Labeling loss ($\mathcal{L}_{\text{SL}}$). This loss minimizes the *cross-entropy* across all the steps as follows:

$$\mathcal{L}_{\text{SL}}(\bar{\pi}) = -\frac{1}{|O|}\sum_{t=1}^{|O|} \log f_\theta(y_t\,|\,\bar{\pi}_{<t}, G), \tag{5}$$

where $y_t \in J$ is the index of the job selected at the decision step $t$ while constructing the solution $\bar{\pi}$.

The rationale behind this training schema is to collect knowledge about sequences of decisions leading to high-quality solutions and distill it in the parameters $\theta$ of the model. This is achieved by treating the best-generated solution $\bar{\pi}$ for an instance $G$ as a pseudo-label and maximizing its likelihood using Eq. 5. Similar to supervised learning, repeated exposure to various training instances enables the model to progressively refine its ability to solve combinatorial problems such as the JSP. Hence, we named this training strategy *Self-Labeling Improvement Method (SLIM)*.

**Relations with Existing Works**

Although our strategy was developed independently, attentive readers may find a resemblance with the Cross-Entropy Method (CEM), a stochastic and derivative-free optimization method [15]. Typically applied to optimize parametric models such as Bernoulli and Gaussian mixtures models [57], the

CEM relies on a maximum likelihood approach to either estimate a random variable or optimize the objective function of a problem [7]. According to Algorithm 2.2 in [7], the CEM independently samples $N = \beta$ solutions, selects a subset based on the problem's objective ($\hat{\gamma} = S_{(N)}$ in our case), and updates the parameters of the model.

While the CEM independently tackles instances by optimizing a separate model for each, SLIM trains a single model on multiple instances to globally learn how to solve a combinatorial problem. Moreover, we adopt a more complex parametric model instead of mixture models, always select a single solution to update the model, and resort to the gradient descent for updating parameters $\theta$. Notably, our strategy also differs from CEM applications to model-based RL, e.g., [57], as we do not rely on rewards in Eq. 5. In summary, our strategy shares the idea of sampling and selecting with the CEM but globally operates as a supervised learning paradigm once the target solution is identified.

Other self-labeling approaches can be found, e.g., in [10, 2], where the former uses K-Means to generate pseudo-labels, and the latter assigns labels to equally partition data with an optimal transportation problem. As highlighted in [2], these approaches may incur in degenerate solutions that trivially minimize Eq. 5, such as producing the same solution despite the input instance in our case. We remark that SLIM avoids such degenerate solutions by jointly using a probabilistic generation process and the objective value (makespan) of solutions to select the best pseudo-label $\overline{\pi}$.

# 5    Results

This section outlines the implementation details, introduces the selected competitors, and presents the key results. All the experiments were performed on an Ubuntu 22.04 machine equipped with an Intel Core i9-11900K and an NVIDIA GeForce RTX 3090 GPU having 24 GB of memory.[1]

## 5.1    Experimental Setup

**Dataset & Benchmarks.** To train our model, we created a dataset of $30\,000$ instances as in [46] by randomly generating 5000 instances per shape ($n \times m$) in the set: $\{10 \times 10, 15 \times 10, 15 \times 15, 20 \times 10, 20 \times 15, 20 \times 20\}$. While our training strategy does not strictly require a fixed dataset, we prefer using it to enhance reproducibility. For testing purposes, we adopted two challenging and popular benchmark sets to evaluate our model and favor cross-comparison. The first set is from Taillard [46], containing 80 instances of medium-large shapes (10 instances per shape). The second set is the Demirkol's one [16], containing 80 instances (10 per shape) which proved particularly challenging in related works [54, 25]. We additionally consider the smaller and easier Lawrence's benchmark [44] and extremely larger instances in Apps. C and E, respectively.

**Metric.** We assess performance on each benchmark instance using the Percentage Gap: $\mathrm{PG} = 100 \cdot (C_{alg}, /, C_{ub} - 1)$, where $C_{alg}$ is the makespan produced by an algorithm and $C_{ub}$ is either the optimal or best-known makespan for the instance. Lower PG values indicate better results, as they reflect solutions with an objective value closer to the optimal or best-known makespan.

**Architecture.** In all our experiments, we configure the model of Sec. 4.1 in the same way. Our *encoder* consists of two GAT layers [8], both with 3 attention heads and leaky slope at 0.15. In $\mathrm{GAT}_1$, we set the size of each head to 64 and concatenate their outputs; while in $\mathrm{GAT}_2$, we increase the head's size to 128 and average their output to produce $e_i \in \mathbb{R}^{143}$ ($h = 15 + 128$). Inside the *decoder's memory network*, the MHA layer follows [49] but it concatenates the output of 3 heads with 64 neurons each, while $W_1 \in \mathbb{R}^{11 \times 192}$ and $W_2 \in \mathbb{R}^{192 \times 128}$ use 192 and 128 neurons, respectively. Thus, the job states $s_j \in \mathbb{R}^d$ have size $d = 128$. Finally, the *classifier* FNN features a dense layer with 128 neurons activated through the Leaky-ReLU (slope $= 0.15$) and a final linear with 1 neuron.

**Training.** We train this generative model with SLIM (see Sec. 4.2) on our dataset for 20 epochs, utilizing the Adam optimizer [19] with a constant learning rate of 0.0002. In each training step, we accumulate gradients over a batch of size 16, meaning that we update the model parameters $\theta$ after processing 16 instances. During training and validation, we fix the number of sampled solutions $\beta$ to 256 and save the parameters producing the lower average makespan on a hold-out set comprising 100 random instances per shape included in our dataset. Training in this way takes approximately 120 hours, with each epoch lasting around 6 hours.

---

[1] Our code is available at: https://github.com/AndreaCorsini1/SelfLabelingJobShop

Table 2: The average PG of the algorithms on the benchmarks. In each row, we highlight in **blue** (**bold**) the best constructive (non-constructive) gap. Shapes marked with * are larger than those seen in training by our GM.

| | | Greedy Constructives | | | | | | | Multiple (Randomized) Constructives | | | | | | | | Non-constructive Approaches | | | | |
| | | PDRs | | | RL | | Our | | PDRs$_{\beta=128}$ | | | RL$_{\beta=128}$ | | | Our | | | | | | |
| | Shape | SPT | MWR | MOR | INSA | L2D | CL | GM | SPT | MWR | MOR | SBH | L2D | CL | GM$_{128}$ | GM$_{512}$ | L2S$_{500}$ | NLS$_A$ | L2S$_{5k}$ | MIP | CP |
|---|---|---|---|---|---|---|---|---|---|---|---|---|---|---|---|---|---|---|---|---|---|
| Taillard's benchmark | 15 × 15 | 26.1 | 19.1 | 20.4 | 14.4 | 26.0 | 14.3 | 13.8 | 13.5 | 13.5 | 12.5 | 11.6 | 17.1 | 9.0 | 7.2 | **6.5** | 9.3 | 7.7 | 6.2 | **0.1** | **0.1** |
| | 20 × 15 | 32.3 | 23.3 | 24.9 | 18.9 | 30.0 | 16.5 | 15.0 | 18.4 | 17.2 | 16.4 | 10.9 | 23.7 | 10.6 | 9.3 | **8.8** | 11.6 | 12.2 | 8.3 | 3.2 | **0.2** |
| | 20 × 20 | 28.3 | 21.8 | 22.9 | 17.3 | 31.6 | 17.3 | 15.2 | 16.7 | 15.6 | 14.7 | 13.6 | 22.6 | 10.9 | 10.0 | **9.0** | 12.4 | 11.5 | 9.0 | 2.9 | **0.7** |
| | 30 × 15* | 35.0 | 24.1 | 22.9 | 21.1 | 33.0 | 18.5 | 17.1 | 23.2 | 19.0 | 17.3 | 15.1 | 24.4 | 14.0 | 11.0 | **10.6** | 14.7 | 14.1 | 9.0 | 10.7 | **2.1** |
| | 30 × 20* | 33.4 | 24.8 | 26.8 | 22.6 | 33.6 | 21.5 | 18.5 | 23.6 | 19.9 | 20.4 | 17.7 | 28.4 | 16.1 | 13.4 | **12.7** | 17.5 | 16.4 | 12.6 | 13.2 | **2.8** |
| | 50 × 15* | 24.0 | 16.4 | 17.6 | 15.9 | 22.4 | 12.2 | 10.1 | 14.1 | 13.5 | 13.5 | 13.2 | 17.1 | 9.3 | 5.5 | **4.9** | 11.0 | 11.0 | 4.6 | 12.2 | **3.0** |
| | 50 × 20* | 25.6 | 17.8 | 16.8 | 20.3 | 26.5 | 13.2 | 11.6 | 17.6 | 14.6 | 14.0 | 20.8 | 20.4 | 9.9 | 8.4 | **7.6** | 13.0 | 11.2 | 6.5 | 13.6 | **2.8** |
| | 100 × 20* | 14.0 | 8.3 | 8.7 | 13.5 | 13.6 | 5.9 | 5.8 | 10.4 | 7.0 | 7.1 | 15.6 | 13.3 | 4.0 | 2.3 | **2.1** | 7.9 | 5.9 | **3.0** | 11.0 | 3.9 |
| | Avg | 27.4 | 19.5 | 20.1 | 18.0 | 27.1 | 14.9 | 13.4 | 17.2 | 15.0 | 14.5 | 14.8 | 20.5 | 10.5 | 8.4 | **7.8** | 12.2 | 11.3 | 7.4 | 8.4 | **2.0** |
| Demirkol's benchmark | 20 × 15 | 27.9 | 27.5 | 30.7 | 24.2 | 39.0 | - | 18.0 | 17.2 | 22.3 | 23.8 | 12.5 | 29.3 | 19.4 | 12.0 | **11.3** | - | - | - | 5.3 | **1.8** |
| | 20 × 20 | 33.2 | 26.8 | 26.3 | 21.3 | 37.7 | - | 19.4 | 18.8 | 18.9 | 21.6 | 13.5 | 27.1 | 16.0 | 13.5 | **12.3** | - | - | - | 4.7 | **1.9** |
| | 30 × 15* | 31.2 | 31.9 | 36.9 | 26.5 | 42.0 | - | 21.8 | 20.7 | 26.8 | 30.7 | 18.2 | 34.0 | 16.5 | 14.4 | **14.0** | - | - | - | 14.2 | **2.5** |
| | 30 × 20* | 34.4 | 32.1 | 32.3 | 28.5 | 39.7 | - | 25.7 | 23.3 | 26.1 | 28.3 | 17.3 | 33.6 | 20.2 | 17.1 | **15.8** | - | - | - | 16.7 | **4.4** |
| | 40 × 15* | 25.3 | 27.0 | 35.8 | 24.7 | 35.6 | - | 17.5 | 17.9 | 23.2 | 30.2 | 14.4 | 31.5 | 17.6 | 11.7 | **10.9** | - | - | - | 16.3 | **4.1** |
| | 40 × 20* | 33.8 | 32.3 | 35.9 | 29.4 | 39.6 | - | 22.2 | 24.5 | 27.6 | 31.8 | 20.3 | 35.8 | 25.6 | 16.0 | **14.8** | - | - | - | 22.5 | **4.6** |
| | 50 × 15* | 24.3 | 27.6 | 34.9 | 23.0 | 36.5 | - | 15.7 | 17.7 | 24.1 | 31.0 | 18.2 | 32.7 | 21.7 | 11.2 | **10.6** | - | - | - | 14.9 | **3.8** |
| | 50 × 20* | 30.0 | 30.3 | 36.8 | 30.2 | 39.5 | - | 22.4 | 23.5 | 26.8 | 32.7 | 22.8 | 36.1 | 15.2 | 15.8 | **15.0** | - | - | - | 22.5 | **4.8** |
| | Avg | 30.0 | 29.4 | 33.7 | 26.0 | 38.7 | - | 20.3 | 20.4 | 24.5 | 28.8 | 17.2 | 32.5 | 19.0 | 14.0 | **13.1** | - | - | - | 14.6 | **3.5** |

**Competitors.** We compare the effectiveness of our methodology against various types of conventional algorithms and ML competitors reviewed in Sec. 2. We divide competitors as follows:

- *Greedy Constructives* generate a single solution for any input instance. We consider two cornerstone ML works for the JSP: the actor-critic (L2D) of [54] and the recent curriculum approach (CL) of [25]. We also coded the INSertion Algorithm (INSA) of [36] and three standard dispatching rules that prioritize jobs based on the Shortest Processing Time (SPT); Most Work Remaining (MWR); and Most Operation Remaining (MOR).

- *Multiple (Randomized) Constructives* generate multiple solutions for an instance by introducing a controlled randomization in the selection process, a simple technique for enhancing constructive algorithms [50, 25]. We consider randomized results of CL and L2D. As only greedy results were disclosed for L2D, we used the open-source code to sample randomized solutions as described in Sec. 4.1. We also consider the three dispatching rules above, randomized by arbitrarily scheduling an operation among the three with higher priority, and the Shifting Bottleneck Heuristic (SBH) [1] that creates multiple solutions while optimizing. All these approaches were seeded with 12345, generate $\beta = 128$ solutions, and return the one with minimum makespan.

- *Non-constructive Approaches* do not rely on a pure constructive strategy for creating JSP solutions. We include two RL-enhanced metaheuristics: the NLS$_A$ in [18] (200 iterations) and the recent L2S proposal of [55] visiting 500 (L2S$_{500}$) and 5000 (L2S$_{5k}$) solutions. We also consider two state-of-the-art solvers for the JSP: Gurobi 9.5 (MIP) solving the disjunctive formulation of [29] and the CP-Sat (CP) of Google OR tools 9.8, both executing with a time limit of 3600 seconds.

For lack of space, we compare our methodology against other learning proposals in App. B.

### 5.2 Performance on Benchmarks

This section evaluates the performance of our generative Graph Model (GM), configured and trained with SLIM as explained in Sec. 5.1. When contrasted with greedy approaches, our GM generates a single solution by picking the job with the highest probability. In the other comparisons, we randomly sample 128 (GM$_{128}$) and 512 (GM$_{512}$) solutions as explained in Sec. 4.1. Note that we coded our GM, PDRs, INSA, SBH, MIP, and CP; while we reported results from original papers of all the ML competitors but L2D, for which we used the open-source code to generate randomized solutions.

Tab. 2 presents the comparison of algorithms on Taillard's and Demirkol's benchmarks, each arranged in a distinct horizontal section. The table is vertically divided into *Greedy Constructive*, *Multiple (Randomized) Constructive*, and *Non-constructive Approaches*; with the results of algorithms categorized accordingly. Each row reports the average PG (the lower the better) on a specific instance shape while the last row (Avg) reports the average gap across all instances, regardless of their shapes.

This table shows that our GM and CL produce lower gaps than PDRs, INSA, and SBH in both the greedy and randomized cases, proving the superiority of neural constructive approaches. Surprisingly,

| Benchmark | RL | | Self-Labeling | | |
|---|---|---|---|---|---|
| | $CL_{UCL}$ | CL | $CL_{SL}$ | $FNN_{SL}$ | GM |
| Taillard | 17.3 | 10.5 | 10.7 | 9.5 | 8.4 |
| Demirkol | - | 19.0 | 20.9 | 14.9 | 14.0 |

Table 3: The average gaps when sampling 128 solutions from architectures trained without and with self-labeling. $CL_{UCL}$ is the model obtained in [25] by training with reward-to-go on random instance shapes (no curriculum learning) and CL is similarly obtained by applying curriculum learning.

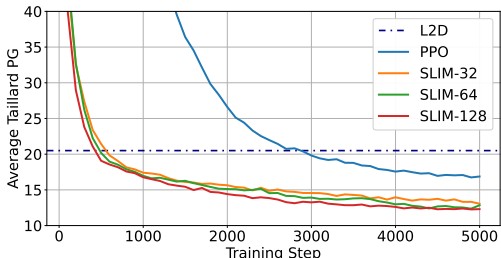

Figure 2: GM validation curves when trained with PPO and our SLIM in the same training setting of [54].

INSA and PDRs outperform L2D. This is likely related to how PDRs were coded in [54], where ours align with those in [25]. Focusing on our GM, we observe that it consistently archives lower gaps than all the greedy approaches and, when applied in a randomized manner ($GM_{128}$), it outperforms all the randomized constructives. Notably, the GM generalizes well to instance shapes larger than training ones (marked with * in Tab. 2), as its gaps do not progressively increase on such shapes.

Our GM is also competitive when compared with non-constructive approaches. The $GM_{512}$ largely outperforms the $NLS_A$ and $L2S_{500}$ metaheuristics, and roughly align with $L2S_{5k}$ visiting 5000 solutions. On shapes larger than $20 \times 20$, the $GM_{512}$ remarkably achieves lower gaps with just a few seconds of computations than a MIP executing for 3600 sec. Therefore, we conclude that our methodology, encompassing the proposed GM and SLIM, is effective in solving the JSP.

As already observed in [25, 55], our GM and other neural constructive approaches are generally outperformed by CP and state-of-the-art metaheuristics, e.g., [36]. Although the performance gap is steadily narrowing, the higher complexity of CP solvers renders them more powerful, albeit at the cost of longer execution time (often dozens of minutes). Therefore, neural constructive approaches may be preferable whenever a quality solution must be provided in a few minutes.

We refer the reader to Apps. D and E for statistical considerations and a comparison between our GM and CP on extremely large instances, respectively.

# 6 A Closer Look at SLIM

This section studies additional aspects of our proposed methodology. For these evaluations, we adopt the same setting described in Sec. 5.1 and explicitly indicate the varied aspects and parameters.

## 6.1 Self-Labeling other Architectures

To demonstrate the generality of our self-labeling improvement method beyond the proposed Graph Model (GM), we use it to train other architectures as described in Sec. 5.1. Specifically, we train the model proposed in [25] ($CL_{SL}$) and a variation of our GM (named $FNN_{SL}$), where we replaced the Graph Attention and Multi-Head Attention layers with linear projections (ReLU activated) by maintaining the same hidden dimensionalities. Tab. 3 reports the average gap of $CL_{SL}$, $FNN_{SL}$, and GM on each benchmark. As baselines for comparison, we also include the average gap of CL and $CL_{UCL}$ reported in [25], the latter being trained without curriculum learning similarly to our setting.

Tab. 3 shows that $CL_{SL}$ nearly matches the performance of CL even without curriculum learning. Note that we intentionally avoided applying curriculum learning to eliminate potential biases when demonstrating the contribution of our self-labeling strategy. Furthermore, we observe that both $FNN_{SL}$ and GM outperform CL, with GM achieving the best overall performance. Therefore, we conclude that SLIM can successfully train well-designed architectures for the JSP.

## 6.2 Comparison with Proximal Policy Optimization

The previous Sec. 6.1 proved the quality of our GM when trained with SLIM. As Proximal Policy Optimization (PPO) is extensively used to train neural algorithms for scheduling problems (see Sec. 2), we contrast the GM's performance when trained with PPO and our self-labeling strategy (SLIM-$\beta$ with $\beta \in \{32, 64, 128\}$). For this evaluation, we adopt the same hyper-parameters of [54] for both

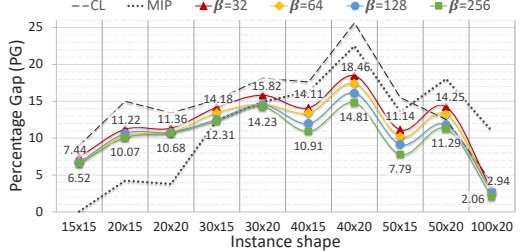 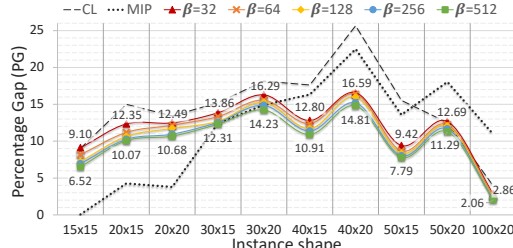

Figure 3: The GM performance when trained by sampling varying number of solutions $\beta$. For each shape, we report the average PG on instances of both benchmarks by sampling 512 solutions during testing.

Figure 4: The GM performance (trained as in Sec. 5.1) for varying numbers of sampled solutions $\beta$ at test time. For each shape, we report the average PG on instances of both benchmarks.

PPO and SLIM by training on $40\,000$ random instances of shape $30 \times 20$ (same training of L2D [54]). We only double the batch size to 8 to reduce training noise. Fig. 2 reports the validation curves obtained by testing the GM on Taillard's benchmark every 100 training steps. We test by sampling 128 solutions and we also include the randomized results of L2D (dashed line) as a baseline for comparisons. From Fig. 2, we observe that training with our self-labeling improvement method results in faster convergence and produces better final models.

We justify this by hypothesizing that the reward received from partial solutions (makespan increments) may not always provide reliable guidance on how to construct the best final solution. While constructing a solution, the critical path may change with implications on past rewards. Our self-labeling strategy does not rely on partial rewards and may avoid such a source of additional "noise". Note that this is only an intuition, proving it would require a deeper analysis, which we refer to in future works. Finally, we stress that we do not claim SLIM is superior to PPO. Instead, we believe it offers an alternative that provides a fresh perspective and potential for integration with existing RL algorithms to advance the neural combinatorial optimization field.

### 6.3 The Effect of $\beta$ on Training

As our self-labeling improvement method is based on sampling, we assess the impact of sampling a different number of solutions $\beta$ while training. We retrain a new GM as described in Sec. 5.1 with a number of solutions $\beta \in \{32, 64, 128, 256\}$, where we stop at 256 as the memory usage with larger values becomes impractical. We test the resulting models by sampling 512 solutions on all the instances of both benchmarks for a broader assessment. Fig. 3 reports the average PG (the lower the better) of the trained GM (colored markers) on each shape. To ease comparisons, we also include the results of CL (dashed line) – the second-best ML proposal in Tab. 2 – and the MIP (dotted line).

Overall, we see that training by sampling more solutions slightly improves the model's performance, as outlined by lower PGs for increasing $\beta$. We also observe that such improvement is less marked on shapes seen in training, such as in $15 \times 15$, $20 \times 15$, and $20 \times 20$ shapes, and more marked on others. This suggests that training by sampling more solutions results in better generalization, although it is more memory-demanding. However, the GM's trends observed in Tab. 2 remain consistent with smaller $\beta$, proving the robustness of SLIM to variations in the number of sampled solutions $\beta$.

### 6.4 The Effect of $\beta$ on Testing

We also assess how the number of sampled solutions $\beta$ impacts the GM performance at test time. To evaluate such an impact, we plot in Fig. 4 the average PG on different shapes for varying $\beta \in \{32, 64, 128, 256, 512\}$. For this analysis, we use the GM trained by sampling 256 solutions, the one of Tab. 2. As in Sec. 6.3, we also report the results of CL and MIP to ease the comparison.

Despite the reduced number of solutions, the GM remains a better alternative than CL – the best RL proposal – and still outperforms the MIP on medium and large instances. Not surprisingly, by sampling more solutions the GM performance keeps improving at the cost of increased execution times. Although we verified that sampling more than 512 solutions further improves results (see bottom of App. F), we decided to stop at $\beta = 512$ as a good trade-off between performance and time. We refer the reader to App. F for other timing considerations.

## 6.5 Self-Labeling the Traveling Salesman Problem

Finally, we provide a brief analysis to demonstrate the broader applicability of our self-labeling improvement method to other combinatorial problems. Thus, we evaluate SLIM on the Traveling Salesman Problem (TSP), a cornerstone in neural combinatorial optimization research [5, 50, 35], by using it to train the well-known Attention Model [27] on TSP instances with 20 nodes. To assess SLIM's performance, we compare it against the established Policy Optimization with Multiple Optima (POMO) approach [30]. We train the Attention Model with both SLIM and POMO, using the same hyperparameters and sampling strategy outlined in [30]. Training is carried out over 100,000 steps (equivalent to 1 epoch in [30]) by generating batches of 32 instances at each step.

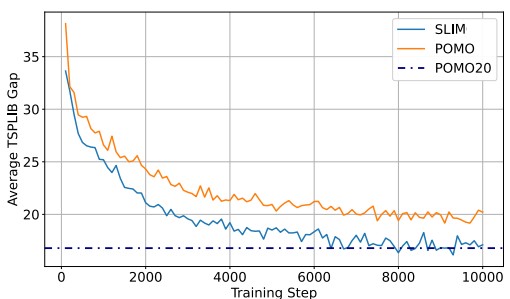

Figure 5: Validation curves obtained by training with SLIM and POMO on random TSP instances with 20 nodes. POMO20 is the best model produced in [30], trained on instances with 20 nodes.

Fig. 5 presents the validation curves of SLIM and POMO obtained by testing the model every 100 steps on small instances (with up to 100 nodes) from the TSPLIB [41]. The results are reported in terms of the average optimality gap and, as a baseline for comparison, we include the performance of the best model in [30] (POMO20) trained for hundreds of epochs. As shown, SLIM achieves faster convergence than POMO and produces a model comparable to POMO20 after just one epoch, thereby demonstrating that SLIM can be effective in other combinatorial problems.

## 7 Limitations

Despite the proven effectiveness of SLIM, it is important to note that only one of the sampled solutions per training instance is used to update the model. This approach may be sub-optimal from an efficiency standpoint. Therefore, we see significant potential in hybridizing our self-labeling strategy with existing (RL) methods to mitigate this inefficiency. Moreover, sampling multiple solutions during training requires substantial memory, which can limit batch sizes. Although we have shown that SLIM remains effective with a small number of sampled solutions (see Sec. 6.3), increasing their number accelerates training convergence and improves the resulting model. Consequently, developing new strategies that can sample higher-quality solutions without generating numerous random ones is a promising future direction. Such advancements could reduce memory usage and further enhance our methodology as well as others in the literature.

## 8 Conclusions

The key contribution of this work is the introduction of SLIM, a novel Self-Labeling Improvement Method to train generative models for the JSP and other combinatorial problems. Additionally, an efficient encoder-decoder architecture is presented to rapidly generate parallel solutions for the JSP. Despite its simplicity, our methodology significantly outperformed many constructive and learning algorithms for the JSP, even surpassing a powerful MIP solver. However, as a constructive approach, it still lags behind state-of-the-art approaches like constraint programming solvers, which nevertheless require more time. We also proved the robustness of SLIM across various parameters and architectures, and its generality by applying it successfully to the Traveling Salesman Problem.

More broadly, self-labeling might be a valuable training strategy as it eliminates the need for optimality information or the precise formulation of a Markov Decision Process. For instance, given a designed generative model (i.e., a constructive algorithm whose decisions are taken by a neural network), this strategy can be applied as-is to combinatorial problems with unconventional objectives or a combination of objective functions. In contrast, RL approaches require the careful definition of a meaningful reward function, which is often a complex and challenging task.

In future, we intend to apply our methodology to other shop scheduling and combinatorial problems as well as explore combinations of our self-supervised strategy with other existing learning approaches.

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

Table 4: The features $x_i \in \mathbb{R}^{15}$ associated with a vertex $i \in V$ of the disjunctive graph $G$ that provide information about operation $i$ in the instance. Recall that $\tau_i$ represents the processing time of operation $i$, and $\mu_i$ denotes the machine on which the operation is performed.

| ID | Description |
|---|---|
| 1 | The processing time $\tau_i$ of the operation. |
| 2 | The completion of job $j$ up to $i$: $\sum_{b=l_j}^{i} \tau_b / \sum_{b \in O_j} \tau_b$. |
| 3 | The remainder of job $j$ after $i$: $\sum_{b=i+1}^{l_j+m_j-1} \tau_b / \sum_{b \in O_j} \tau_b$. |
| 4-6 | The $1^{st}$, $2^{nd}$, and $3^{rd}$ quartile among processing times of operations on job $j$. |
| 7-9 | The $1^{st}$, $2^{nd}$, and $3^{rd}$ quartile among processing times of operations on machine $\mu_i$. |
| 10-12 | The difference between $\tau_i$ and the $1^{st}$, $2^{nd}$, and $3^{rd}$ quartile among processing times of operations in job $j$. |
| 13-15 | The difference between $\tau_i$ and the $1^{st}$, $2^{nd}$, and $3^{rd}$ quartile among processing times of operations on machine $\mu_i$. |

Table 5: The average PG of greedy constructive approaches on the same two Taillard's instances of the ten available for each shape, as utilized in [20, 11]. We highlight in **bold** the best PG on each instance shape.

| | PDRs | | | | RL | | | | |
|---|---|---|---|---|---|---|---|---|---|
| Shape | SPT | MWR | MOR | INSA | TRL | L2D | CL | DQN | GM |
| $15 \times 15$ | 17.5 | 18.8 | 15.2 | 15.5 | 35.4 | 22.3 | 15.2 | **7.1** | 15.1 |
| $20 \times 15$ | 29.7 | 24.5 | 26.8 | 15.3 | 32.1 | 26.8 | 17.9 | **13.9** | 14.0 |
| $20 \times 20$ | 27.8 | 22.0 | 23.1 | 16.7 | 28.3 | 30.3 | **16.2** | 17.9 | 17.4 |
| $30 \times 15^*$ | 39.5 | 22.8 | 23.9 | 21.1 | 36.4 | 31.5 | 20.3 | **16.1** | 17.0 |
| $30 \times 20^*$ | 30.0 | 27.7 | 27.5 | **19.2** | 34.7 | 38.3 | 22.15 | 21.8 | 21.1 |
| $50 \times 15^*$ | 30.1 | 23.8 | 26.2 | 12.4 | 31.85 | 24.6 | 15.6 | 17.7 | **11.8** |
| $50 \times 20^*$ | 24.3 | 18.4 | 18.6 | 23.0 | 28.03 | 28.6 | **14.2** | 19.5 | 14.4 |
| $100 \times 20^*$ | 14.2 | 8.2 | 8.5 | 15.2 | 18.0 | 12.5 | 5.5 | 9.5 | **5.0** |
| Avg | 26.6 | 20.8 | 21.2 | 17.3 | 30.6 | 26.9 | 15.9 | 15.4 | **14.5** |

# Appendix

## A Disjunctive Graph Features

Following ML works reviewed in Sec. 2, we augment the standard disjunctive graph representation of a JSP instance by incorporating additional features for each vertex $i \in V$. Tab. 4 details these features. Some features, such as features 1-3, have been adapted from previous works [54, 39], while others have been introduced by us to better model JSP concepts like machines and jobs. For instance, features 4-6 take the same value for operations of the same job, emphasizing which operations belong to different jobs, while features 7-9 do the same for operations to be performed on the same machine.

Note that these features as well as those outlined in Tab. 1 can also be used for other shop scheduling problems, including Flow Shop, Flexible Flow Shop, and Flexible Job Shop.

## B Additional Comparisons with Neural Algorithms

We additionally consider learning proposals reviewed in Sec. 2, which were tested only on a few benchmark instances or which could not be included in Tab. 2 due to how results are disclosed.

Regarding proposals not included in Tab. 2, we highlight that our GM outperforms the Graph Neural Network in [39] that exhibits a significantly higher average gap of $19.5\%$ on Taillard's benchmark. Note that only qualitative results are disclosed in [39], we did our best to report them. Similarly, we remark that the GM performance is superior to the proposal in [47] (hCP), mixing RL and constraint programming. hCP achieves an average makespan of 2670 and 5701 on Taillard and Demirkol benchmarks respectively, while our greedy GM yet obtains an average of 2642 and 5581.

As some other learning proposals like [11, 20] were tested only on certain benchmark instances, we provide an additional evaluation on such instances for a fair comparison. We include in Tab. 5 the average PGs of greedy constructive approaches on the two Taillard's instances of each shape used in the Transformer (TRL) proposal of [11] and the Deep Q-Network (DQN) of [20]. On this subset of

Table 6: The average PGs of the algorithms on Lawrence's benchmark. For each row, we highlight in **blue** and **bold** the lowest (best) constructive and non-constructive gap, respectively.

| Shape | Greedy Constructives | | | | | | Multiple (Randomized) Constructives | | | | | | | Non-constructive Approaches | | | |
| | PDRs | | | | RL | Our | PDRs ($\beta = 128$) | | | | RL | Our | | | | | |
| | SPT | MWR | MOR | INSA | L2D | GM | SPT | MWR | MOR | SBH | L2D | GM$_{128}$ | GM$_{512}$ | L2S$_{500}$ | L2S$_{5k}$ | MIP | CP |
|---|---|---|---|---|---|---|---|---|---|---|---|---|---|---|---|---|---|
| $10 \times 5$ | 14.8 | 15.9 | 14.6 | 7.3 | 14.3 | 9.3 | 3.8 | 9.0 | 5.6 | 2.0 | 8.8 | 1.9 | **1.1** | 2.1 | 1.8 | **0.0** | **0.0** |
| $15 \times 5$ | 14.9 | 5.5 | 5.0 | 2.2 | 7.8 | 2.6 | 2.3 | 1.2 | 1.1 | **0.0** | 2.8 | **0.0** | **0.0** | **0.0** | **0.0** | **0.0** | **0.0** |
| $20 \times 5$ | 16.3 | 5.2 | 6.7 | 2.7 | 6.3 | 2.1 | 5.8 | 1.4 | 2.6 | 0.1 | 3.1 | **0.0** | **0.0** | **0.0** | **0.0** | **0.0** | **0.0** |
| $10 \times 10$ | 15.7 | 12.2 | 12.0 | 7.9 | 23.7 | 8.9 | 8.5 | 7.6 | 6.9 | 5.7 | 10.4 | 3.1 | **2.5** | 4.4 | 0.9 | **0.0** | **0.0** |
| $15 \times 10$ | 28.7 | 17.8 | 23.4 | 12.1 | 27.2 | 11.6 | 15.2 | 11.1 | 11.6 | 7.6 | 16.2 | 5.2 | **5.0** | 6.4 | 3.4 | **0.0** | **0.0** |
| $20 \times 10$ | 36.9 | 17.9 | 21.9 | 16.0 | 24.6 | 12.1 | 17.7 | 12.5 | 12.9 | 5.8 | 18.3 | 6.9 | **5.6** | 7.0 | 2.6 | **0.0** | **0.0** |
| $30 \times 10$ | 16.6 | 9.2 | 7.7 | 3.9 | 8.4 | 2.0 | 8.5 | 3.5 | 2.3 | **0.0** | 6.8 | **0.0** | **0.0** | 0.2 | **0.0** | **0.0** | **0.0** |
| $15 \times 15$ | 25.8 | 18.2 | 18.7 | 14.8 | 27.1 | 13.6 | 12.3 | 12.2 | 12.8 | 8.4 | 17.4 | 6.8 | **5.6** | 7.3 | 5.9 | **0.0** | **0.0** |
| Avg | 21.2 | 12.7 | 13.7 | 8.4 | 17.4 | 7.8 | 9.3 | 7.3 | 7.0 | 3.7 | 10.6 | 3.0 | **2.5** | 3.4 | 1.8 | **0.0** | **0.0** |

Table 7: The average PG and its standard deviation ($avg \pm std$) of the best four multiple (randomized) constructive and non-constructive algorithms on Taillard's benchmark. We highlight in **blue** (**bold**) the best constructive (non-constructive) gap.

| Shape | Multiple (Randomized) Constructives | | | | Non-constructive Approaches | | | |
| | MOR | CL | GM$_{128}$ | GM$_{512}$ | NLS$_A$ | L2S$_{5k}$ | MIP | CP |
|---|---|---|---|---|---|---|---|---|
| $15 \times 15$ | $12.5 \pm 2.6$ | $9.0 \pm 2.0$ | $7.2 \pm 1.6$ | $\mathbf{6.5 \pm 1.3}$ | $7.7 \pm 3.0$ | $6.2 \pm 1.4$ | $0.1 \pm 0.2$ | $\mathbf{0.1 \pm 0.1}$ |
| $20 \times 15$ | $16.4 \pm 1.5$ | $10.6 \pm 1.6$ | $9.3 \pm 1.8$ | $\mathbf{8.8 \pm 1.4}$ | $12.2 \pm 1.7$ | $8.3 \pm 1.7$ | $3.2 \pm 1.9$ | $\mathbf{0.2 \pm 0.4}$ |
| $20 \times 20$ | $14.7 \pm 1.7$ | $10.9 \pm 1.6$ | $10.0 \pm 1.3$ | $\mathbf{9.0 \pm 1.0}$ | $11.5 \pm 1.5$ | $9.0 \pm 2.2$ | $2.9 \pm 1.6$ | $\mathbf{0.7 \pm 0.7}$ |
| $30 \times 15$ | $17.3 \pm 4.5$ | $14.0 \pm 3.8$ | $11.0 \pm 3.8$ | $\mathbf{10.6 \pm 3.6}$ | $14.1 \pm 3.5$ | $9.0 \pm 3.0$ | $10.7 \pm 3.1$ | $\mathbf{2.1 \pm 2.2}$ |
| $30 \times 20$ | $20.4 \pm 2.2$ | $16.1 \pm 2.0$ | $13.4 \pm 1.6$ | $\mathbf{12.7 \pm 1.4}$ | $16.4 \pm 2.4$ | $12.6 \pm 2.1$ | $13.2 \pm 2.9$ | $\mathbf{2.8 \pm 1.5}$ |
| $50 \times 15$ | $13.5 \pm 4.5$ | $9.3 \pm 3.8$ | $5.5 \pm 3.0$ | $\mathbf{4.9 \pm 2.7}$ | $11.0 \pm 4.3$ | $4.6 \pm 3.1$ | $12.3 \pm 3.9$ | $\mathbf{3.0 \pm 0.0}$ |
| $50 \times 20$ | $14.0 \pm 1.5$ | $9.9 \pm 2.2$ | $8.4 \pm 2.0$ | $\mathbf{7.6 \pm 2.0}$ | $11.3 \pm 2.1$ | $6.5 \pm 1.9$ | $13.6 \pm 4.1$ | $\mathbf{2.8 \pm 1.9}$ |
| $100 \times 20$ | $7.1 \pm 2.9$ | $4.0 \pm 2.2$ | $2.3 \pm 1.1$ | $\mathbf{2.1 \pm 1.2}$ | $5.9 \pm 2.3$ | $\mathbf{3.0 \pm 1.7}$ | $11.0 \pm 3.5$ | $3.9 \pm 1.5$ |
| Tot Avg $\pm$ Std | $14.5 \pm 4.6$ | $10.5 \pm 4.2$ | $8.4 \pm 3.8$ | $\mathbf{7.8 \pm 3.6}$ | $11.3 \pm 4.1$ | $7.4 \pm 3.7$ | $8.4 \pm 5.9$ | $\mathbf{2.0 \pm 1.9}$ |

instances, we see that TRL is the worst-performing approach and that DQN roughly aligns with CL. As our GM is always the best or second best algorithm on each shape, it achieves the lowest overall average gap (Avg).

## C   Performance on Lawrence's Benchmark

Lawrence's benchmark has been extensively used in the JSP literature to develop various resolution approaches, e.g. [1, 36, 55]. This benchmark includes 40 instances of 8 different shapes and features smaller (and easier) instances compared to the Taillard's and Demirkol's ones. To complement the evaluation of Sec. 5.2 on small instances, we provide in Tab. 6 a comparison of approaches included in Tab. 2 on Lawrence's benchmark. Note that we could not include CL and NLS$_A$ as they were not tested on these instances.

Tab. 6 shows that our GM remains the best constructive approach for small instances. However, we observe that L2S$_{5k}$ achieves slightly better results than GM$_{512}$ at the cost of significantly longer execution times. Specifically, L2S$_{5k}$ always requires more than 70 sec on such instance shapes (as reported in [55]), whereas our GM$_{512}$ always completes in less than 0.5 sec (see also App. F for timing considerations). Finally, we highlight that on small instances like those in Lawrence's benchmark, exact methods such as MIP and CP can effectively solve them to optimality.

## D   Statistical Analysis

Due to space constraints, we evaluate algorithms in Tab. 2 solely in terms of average gaps. This approach is justified as the average trends are consistent with other statistical measures. For example, Tab. 7 reports the average and standard deviation of the best four multiple (randomized) constructive and non-constructive approaches on Taillard's benchmark. We omit Demirkol's instances since not all algorithms report results for them. Each row in Tab. 7 presents the average gap and standard deviation for instances of a specific shape, but the last row, which reports the average and standard deviation on all instances, regardless of their shapes. This table confirms that the

Table 8: Average makespan ($avg \pm std$) and execution time of CP-Sat and our GM (when sampling $\beta = 512$ solutions) on very large instances.

| | | Instance shape | | | |
|---|---|---|---|---|---|
| | | $200 \times 20$ | $200 \times 40$ | $500 \times 20$ | $500 \times 40$ |
| CP-Sat | $C_{max}$ | $21848.5 \pm 331$ | $23154.6 \pm 350$ | $53572.7 \pm 743$ | $546064.0 \pm 443$ |
| | $time$ | $3600\ sec$ | $3600\ sec$ | $3600\ sec$ | $3600\ sec$ |
| $GM_{512}$ | $C_{max}$ | $21794.5 \pm 324$ | $22795.0 \pm 277$ | $53288.4 \pm 737$ | $53942.9 \pm 425$ |
| | $time$ | $31\ sec$ | $64\ sec$ | $164\ sec$ | $343\ sec$ |
| Gap sum from CP-Sat | | -3.30 | -30.88 | -6.30 | -26.14 |

observations made for Tab. 2 remain valid when considering standard deviations. Specifically, $GM_{512}$ remains the best constructive approach, aligning closely with the RL-enhanced metaheuristic $L2S_{5k}$. The MIP still has worse overall performance than $GM_{512}$ while CP produces the best average gaps and standard deviations.

## E    Extremely Large Instances

As shown in Tab. 2, constructive approaches are generally inferior to state-of-the-art metaheuristics [36] (see also [55]), Mixed Integer Programming (MIP) [29], and Constraint Programming (CP) solvers. These resolution methods employ more complex frameworks, making them more effective for solving combinatorial problems. However, neural constructive approaches are continuously closing the gap as remarked by the GM's performance matching that of MIP solvers. Herein, we also prove that our GM generally produces better results than CP solvers on very large instances.

To this end, we randomly generated 20 JSP instances with shapes $\{200 \times 20, 200 \times 40, 500 \times 20, 500 \times 40\}$ and solved them with the CP-Sat solver of Google OR-tools 9.8, executing for 1 hour, and our GM (trained as described in Sec. 5.1), sampling $\beta = 512$ solutions. Tab. 8 reports the average execution time ($time$), the average makespan and its standard deviation ($C_{max}$) of CP-Sat and our $GM_{512}$ on each shape. The last row (Gap sum from CP-Sat) offers a relative comparison by summing up the gaps of the $GM_{512}$ computed from the solutions produced by CP-Sat. A negative value means that the GM produced better solutions.

As one can see, the $GM_{512}$ produces better average results than CP-Sat with just a fraction of time. On these 80 instances, we observed only 7 cases in which the solutions of CP-Sat are slightly better than those of the GM. This remarks once more the quality of our proposal, making it a valid alternative to CP solvers for extremely large instances.

Lastly, we remark that on these large instances, the execution times of our GM can be further reduced by employing speed-up techniques such as model quantization (i.e., converting floats to lower precision numbers) and the new compile functionality available in PyTorch $\geq 2.0$. Our model did not undergo any of the above (herein and in any other evaluation) and has been coded by keeping the implementation as simple as possible.

## F    Execution Times

We additionally assess the timing factor, an important aspect in some scheduling scenarios. To this end, we plot in the left of Fig. 6 the execution time trends of dispatching rules (PDR), INSA, L2D (we omit CL as it has slightly higher execution times than L2D), and GM on typical benchmark shapes, all applied in a greedy constructive manner. We also include the execution time of our GM when sampling 512 solutions ($GM_{512}$) to prove that sampling multiple parallel solutions does not dramatically increase times. Note that all these algorithms were executed on the machine described at the beginning of Sec. 5. We omit other RL proposals as they do not publicly release the code, do not disclose execution times, or is unclear whether they used GPUs.

When sampling 512 solutions, the $GM_{512}$ takes less than a second on medium shapes, less than 3 sec on large ones, and around 10 sec on the big $100 \times 20$. As these trends align with L2D, which only constructs a single solution, our GM is a much faster neural alternative, despite being also better in terms of quality. We also underline that the GM and any ML-based approach are likely to be

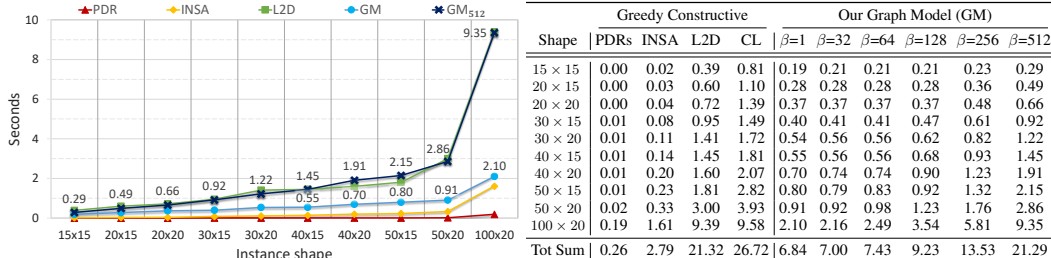

| Shape | Greedy Constructive | | | | Our Graph Model (GM) | | | | | |
|---|---|---|---|---|---|---|---|---|---|---|
| | PDRs | INSA | L2D | CL | $\beta$=1 | $\beta$=32 | $\beta$=64 | $\beta$=128 | $\beta$=256 | $\beta$=512 |
| $15 \times 15$ | 0.00 | 0.02 | 0.39 | 0.81 | 0.19 | 0.21 | 0.21 | 0.21 | 0.23 | 0.29 |
| $20 \times 15$ | 0.00 | 0.03 | 0.60 | 1.10 | 0.28 | 0.28 | 0.28 | 0.28 | 0.36 | 0.49 |
| $20 \times 20$ | 0.00 | 0.04 | 0.72 | 1.39 | 0.37 | 0.37 | 0.37 | 0.37 | 0.48 | 0.66 |
| $30 \times 15$ | 0.01 | 0.08 | 0.95 | 1.49 | 0.40 | 0.41 | 0.41 | 0.47 | 0.61 | 0.92 |
| $30 \times 20$ | 0.01 | 0.11 | 1.41 | 1.72 | 0.54 | 0.56 | 0.56 | 0.62 | 0.82 | 1.22 |
| $40 \times 15$ | 0.01 | 0.14 | 1.45 | 1.81 | 0.55 | 0.56 | 0.56 | 0.68 | 0.93 | 1.45 |
| $40 \times 20$ | 0.01 | 0.20 | 1.60 | 2.07 | 0.70 | 0.74 | 0.74 | 0.90 | 1.23 | 1.91 |
| $50 \times 15$ | 0.01 | 0.23 | 1.81 | 2.82 | 0.80 | 0.79 | 0.83 | 0.92 | 1.32 | 2.15 |
| $50 \times 20$ | 0.02 | 0.33 | 3.00 | 3.93 | 0.91 | 0.92 | 0.98 | 1.23 | 1.76 | 2.86 |
| $100 \times 20$ | 0.19 | 1.61 | 9.39 | 9.58 | 2.10 | 2.16 | 2.49 | 3.54 | 5.81 | 9.35 |
| Tot Sum | 0.26 | 2.79 | 21.32 | 26.72 | 6.84 | 7.00 | 7.43 | 9.23 | 13.53 | 21.29 |

Figure 6: The average execution time in seconds of the coded algorithms on different instance shapes considered in Tab. 2. The times of PDRs, INSA, L2D, and CL refer to the construct of a single solution. Whereas we report the times of the GM for varying numbers of sampled solution $\beta \in \{1, 32, 64, 128, 256, 512\}$.

slower than PDRs and constructive heuristics, either applied in a greedy or randomized way. One can see that such approaches work faster: PDRs take 0.2 sec and INSA 1.61 sec on $100 \times 20$ instances. However, this increase in execution time is not dramatic and is largely justified by better performance, especially in the case of our GM.

We finally provide in the right table of Fig. 6 the precise GM timing for varying numbers of sampled solutions $\beta$ during testing. Each row reports the average execution time on a benchmark shape and the last row (Tot Sum) sums up all the times for an overall comparison. We observe that sampling 32 and 64 solutions from the GM takes the same except in large instances ($50 \times 15, 50 \times 20$, and $100 \times 20$ shapes), where variations are small. Therefore, sampling less than 32 solutions is pointless, as the reduction in the execution times is negligible while the quality is further reduced. Additionally, sampling more than 512 solutions further improves the GM's results at the cost of a consistent increment in the execution time. As an example, by sampling 1024 solutions, the overall average gap (Avg row in Tab. 2) is reduced by 0.4% in both benchmarks with respect to the $\beta = 512$ case, but the execution time is roughly 1.9 times larger. Instead of blindly sampling many random solutions with long executions, it should be possible to achieve better results in less time with ad-hoc strategies, such as Beam-Search or other strategies, that sample differently based on the generated probabilities or other information. We leave this study on testing strategies to future work.

## G  Extended Results

The extended results of the coded algorithms (PDRs, INSA, SBH, L2D, MIP, CP, and the GM's configurations) are available at [2]. For computing the percentage gap (PG), we used the optimal or best-known makespan of an instance ($C_{ub}$), available at: https://optimizizer.com/jobshop.php. The extended results highlight that models trained using our Self-Labeling strategy with $\beta < 256$ exhibit competitive performance and, in some instances, even surpass the model trained with $\beta = 256$. This indicates the efficacy of our strategy even with a reduced number of sampled solutions. However, as remarked in the paper, sampling a larger number of training solutions yields improved overall performance, as evidenced by the lower total sums observed for larger values of $\beta$.

---

[2]Extended results: https://github.com/AndreaCorsini1/SelfLabelingJobShop/blob/main/output/Results.xlsx

