# OpenReview forum: "Self-Labeling the Job Shop Scheduling Problem"
_NeurIPS.cc/2024/Conference — NeurIPS 2024 poster_

### Official Review · Reviewer_mZNb · 2024-07-07

**Soundness:** 3
**Presentation:** 4
**Contribution:** 3
**Rating:** 6
**Confidence:** 3

**Summary:**

This paper is empirical in nature and studies generative models for the job shop scheduling problem (JSP).  JSP is well-studied in the scheduling community, both theoretically and empirically, in part because of its many applications. In JSP, a DAG is given of the precedence ordering of a set of operations (equipped with a specified machine it should be processed on and the amount of machine time it will require), and jobs are sequences of operations that must be completed respecting the order of the precedence relations. The available machines can process one operation at a time, and the goal is to minimize the makespan, i.e. the time when the last job is completed.

The training strategy (known as the self-labeling strategy) is to generate many solutions for an instance, then choose the best one according to the objective to be the ”pseudo-label”. Generating solutions is done by making sequential decisions about which operation per job should be scheduled next on a machine. The architecture for these decisions is called a Pointer Network, which basically makes its choices by learning a function estimating the probability of a solution being of high quality. Then, generating solutions is done via an intuitive sampling procedure.

The experimental sections compares against some baseline ML works for JSP called L2D and CL.  They additionally compare against some classic theoretical heuristics, including shortest processing time SPT and most work remaining MWR. The studies seemed well-constructed. The authors show that their algorithm (plural algorithms I suppose since they consider some different parameters for how many solutions for generated before the best is chosen of the pseudo-label) perform significantly better in terms of the quality in the solution on 2 standard sets of benchmarks. Additional strategies are compared against in the appendix, and the conclusion still stands.

**Strengths:**

I find the paper to be well-motivated. The authors state that while meta-heuristics are state-of-the-art for the JSP, they are very expensive to compute. On the other hand, semi-supervised and self-supervised learning (which can learn from unlabeled data) seem more promising for combinatorial optimization problems, despite this area being understudied so far.

The assumptions for the broad techniques to be useful for other combinatorial optimization problems are rather weak: (1) one must be able to generate multiple feasible solutions to the problem and (2) one must be able to evaluate the objective of said solutions. Such weak assumptions suggest that this framework will likely be useful for a broader range of problems in CO.

The empirical results in this paper indicate that their strategy is better than previous works for JSP, excluding Constraint Programming (CP) solvers and meta-heuristics. Experiments feel complete and well-elaborated upon.

The presentation of the paper is very nice.

**Weaknesses:**

The authors note that while there are techniques that can produce higher quality solutions than their algorithms (Constraint Programming (CP) solvers and meta-heuristics), these seem to be much more computationally expensive techniques, which are not really useful for large instances.

I am unsure how motivating JSP is for generative models, since simple algorithms already perform quite well, i.e. list scheduling. I don’t find it the most motivating scheduling problem for initiating the study of generative models in scheduling. Perhaps a bit niche.

**Questions:**

What specific combinatorial properties does JSP have that made this amenable to your techniques? I ask because while I believe these methods can be extended to some other CO problems, I’m trying to understand what broader class of CO problems your techniques could be effective for.

Do you see any connection between the work in generative models for CO problems and the work on learning-augmented algorithms (also known as algorithms with predictions)?  In particular, is there any reason to believe the problems in CO for which generative models may be useful are the same as the problems that can be improved in the algorithms with predictions framework?

I am unfamiliar with the empirical benchmarks in this area. Is there any reason to fear that Taillard’s,  Demirkol’s benchmarks, and the randomly generated instances have some similarities that are not shared by other JSP?

**Limitations:**

Yes

---

> ### Author Rebuttal · Authors · 2024-08-05
>
> Please find our responses to the main concerns and questions below:
> - **A1 (CP observation):** We agree: metaheuristics and CP require more time to provide quality solutions, especially on large instances. **This is the goal of Appendix D, showing that CP scales worse on very large instances.** Note our objective was to stress that neural constructive approaches still lag behind state-of-the-art methodologies in terms of quality and that further research is required to bridge this gap.
> - **A2 (motivating JSP):** Simple list schedulers are far from performing well as remarked by their large gaps in Tab. 2 of our paper. **The Job Shop is an extensively studied problem with many proposed algorithms, available benchmarks, and practical applications.** Despite this, (neural) constructive heuristics still have consistent limitations, see lines 21-35. **Effectively solving the JSP also allows solving special JSP cases (like Flow Shop) as well as establishing a concrete base for tackling more complicated variants such as Flexible JSP.** Note also the increasing number of neural combinatorial publications considering the JSP. Therefore, Job Shop constitutes a perfect playground for developing new learning-based generative solutions.
> - **A3 (broader impact):** We do not exploit specific JSP properties with our self-labeling strategy, but we do leverage JSP characteristics when designing our model. We focus on the renowned Job Shop problem because neural approaches tend to be less effective on scheduling problems (see also your A2) compared e.g. to routing ones. However, if you have an effective generative model (e.g., a Pointer Network) for a CO problem, you can apply our self-labeling strategy. For instance, there exist recent follow-up works (e.g., Pirnay and Dominik, 2024) using our strategy for routing problems (like TSP and VRP), demonstrating its versatility beyond scheduling. **More broadly, wherever you can apply a constructive algorithm, you can design a generative model and apply our strategy.** Thank you, we will include this consideration as broader impact.
> - **A4 (learning-augmented algorithms):** We apologize, but we are unsure about the meaning of learning-augmented algorithms. If you are referring to metaheuristics enhanced with deep learning, as long as you can have multiple solutions/options (generative model) and discriminate based on an objective, you can apply self-labeling as is. In the case of predictive and prescriptive models, it may still be possible. For instance, one may use the top $\beta$ suggestions of the model, evaluate the responses of the algorithm, and reinforce (one-hot label) the one resulting in the best response. Hope this partially answers you.
> - **A5 (benchmark similarities):** In scheduling, benchmarks contain hard-to-solve instances that have been used throughout the literature. In our work, we use arbitrary generated instances to learn solving the JSP and the benchmarks serve as standard test sets to evaluate the model/algorithm. **There are no particular similarities that must be preserved in training instances, as long as they are JSP instances.** In JSP variants, like Flexible JSP, you can generate arbitrary instances, train on them, and evaluate on benchmarks. While some studies suggest that generating certain types of instances improve learning to solve CP problems, (especially routing ones), this is not what we did.

---

> ### Comment · Reviewer_mZNb · 2024-08-09
>
> Thank you for your response.
>
> I find your point (A3) to be rather motivating, and I see that you included similar responses to the other reviewers, particularly in mentioning TSP, flow shop, and flexible JSP.
> Just so you are aware, the line of work I was talking about is this: https://algorithms-with-predictions.github.io.
>
> This was the main critique of the paper, so I have raised my score from a 5 to a 6.

---

### Official Review · Reviewer_NWh3 · 2024-07-12

**Soundness:** 3
**Presentation:** 3
**Contribution:** 3
**Rating:** 5
**Confidence:** 4

**Summary:**

This paper proposes a job-shop scheduling method based on self-labeling strategy and pointer network. The structure of the paper is clear. The method is evaluated on public benchmarks Taillard and Demirkol’s.

**Strengths:**

Overall speaking, the self labeling strategy is an interesting approach because it only requires an objective function for determining the optimal solution in the current solution set in order to train the model. It avoids the expensive cost of using a solver. Also, this approach is easier to implement than reinforcement learning. The results show that the proposed method achieves better performance than PDRs and RL on two public benchmarks.

**Weaknesses:**

1. The self-labeling strategy necessitates the generation of a large number (denoted as beta) of solutions for each training epoch, with only one of these solutions being valid. Consequently, this method exhibits notably low sample utilization. As depicted in Figure 2, to achieve a well-trained model, beta typically needs to be set at 256 or even higher, resulting in sample utilization well below 1%. Are there any methods or ideas available to enhance sample utilization in this context?

2. While the paper validates the method solely on two public benchmarks (TA and DMU), it's worth noting that there exist additional public benchmarks for JSP, including ABZ, FT, LA, ORB, SWV, and YN [1-6]. Considering these benchmarks could provide a more comprehensive evaluation of the proposed approach.

3. To ensure the reproducibility of the experimental results, it is essential to make the source code publicly available on platforms like GitHub. This transparency is crucial for reviewers to verify the credibility of the results presented in the paper.

[1] J. Adams, E. Balas, and D. Zawack. The shifting bottleneck procedure for job shop scheduling. Management Science, 34.3: 391-401, 1988.

[2] H. Fisher and G. L. Thompson. Probabilistic learning combinations of local job-shop scheduling rules. In: Industrial Scheduling: 225-251. ed. by J.F. Muth and G.L. Thompson. Prentice Hall, 1963.

[3] S. Lawrence. Resource Constrained Project Scheduling. An Experimental Investigation of Heuristic Scheduling Techniques (Supplement). Carnegie-Mellon University, 1984.

[4] D. Applegate and W. Cook. A computational study of job-shop scheduling. ORSA Journal of Computing, 3.2: 149-156, 1991.

[5] R.H. Storer, S.D. Wu and R. Vaccari. New search spaces for sequencing instances with application to job shop scheduling. Management Science, 38.10: 1495-1509, 1992.

[6] T. Yamada and R. Nakano. A genetic algorithm applicable to large-scale job-shop instances. In: Parallel instance solving from nature II: 281-290. ed. by R. Manner and B. Manderick. Elsevier, 1992.

**Questions:**

1. Have you explored the possibility of transitioning your work to a different variant of JSP, like FJSP?
2. The paper appears to lack any discussion regarding the solution time of the proposed method. I am also interested in understanding the solution time of your algorithm and the duration required for its training.
3. The Encoder in the paper operates at the operation level, while the Decoder functions at the job level. What factors influenced this design choice? Have you ever experimented with a network structure that is entirely operation-level or job-level?

**Limitations:**

The paper has discussed some limitations in the appendix.

---

> ### Author Rebuttal · Authors · 2024-08-05
>
> Please find our responses to the main concerns and questions below:
> - **A1 (large $\beta$ and utilization):** Fig. 2 of our paper proves that training the SPN with $\beta = 32$ significantly outperforms CL, the best neural constructive. **Increasing $\beta$ enhances performance, but high $\beta$ values are not needed to outperform baselines (see lines 322-327).** We agree about sample utilization: **in App. F we state that utilization is an efficiency limitation and constitute a potential for future improvements.** For instance, exploring the use of a portion of samples as labels may be beneficial, but is not required for our method effectiveness.
> - **A2 (benchmarks):** You are right. We realized that the evaluation in the main paper does not properly cover small-medium instances (contained in the cited benchmarks). Thus, **we have extended the evaluation on Lawrence's benchmark in the Tab. 1 of the attached PDF file (above).** Remarkably, the SPN remains the best constructive approach and the results show that MIP and CP close all these smaller and easier instances (quite fast). Note also that our experimental evaluation aligns with that of published works cited in the paper. Thus, we are very confident about the quality of our proposal as the SPN was thoroughly tested in challenging scenarios (including very large instances in App. D).
> - **A3 (code):** We totally agree. **The code is available in the supplementary material and also on GitHub** (we did not directly link to GitHub for keeping anonymity).
> - **A4 (FJSP extension):** **We are exploring extensions, including the FJSP.** While elements like the self-labeling strategy and features can be used for the FJSP, the architecture and solution construction process will need careful re-evaluation to ensure effectiveness. We would be happy to collaborate if there is the opportunity.
> - **A5 (execution times):** We point the reviewer to **lines 334-338**, where we consider execution time and refer to Appendix E for detailed considerations. Whereas, in **lines 258 and 259**, we state that training roughly takes 120 hours, around 6 hours per epoch.
> - **A6 (model choices):** Based on the considerations in Sec. 3.1, each job has a single active operation during the solution construction. We structure the decoder's decisions at the job-level to avoid accounting for inactive operations, thereby simplifying the decoder's task. Meanwhile, the encoder captures instance-wide relationships in the operations' embeddings using the disjunctive graph. **This approach allows the encoder to maintain a high-level view of the instance characteristics, while the decoder focuses on the solution construction, specifically the status of machines and jobs.** We did not explore models fully working at the operation or job level, but these are plausible alternatives (see also A3 of reviewer 2iGC).

---

> > ### Comment · Reviewer_NWh3 · 2024-08-13
> >
> > I appreciate the authors for supplementing Lawrence's benchmark to provide additional validation for their work. I also have a question regarding the choice of using Pointer Network over Transformer architecture for the encoding and decoding processes. It appears that Pointer Network is an outdated network structure. In essence, the primary contribution of this study lies in the introduction of a Self-Labeling training strategy elaborated in Section 4.2. However, I find the contribution somewhat limited in terms of its impact on improving my score.

---

> > > ### Author Response · Authors · 2024-08-13
> > > **Pointer Network motivation**
> > >
> > > As remarked by the reviewer, our primary contribution is the Self-Labeling strategy. Due to the volume of works adopting the well-established Pointer Network framework, we adhered to this choice to put the emphasis on the proposed learning methodology. Our architecture was tailored to the JSP by leveraging related works (Sec. 2). It integrates Graph Neural Network layers for encoding the disjunctive graph (as in [47, 37, 10]) and employs Multi-Head Attention with a Feed-Forward network (inspired by Transformers) for scheduling jobs in decoding similarly to [10, 24]. Moreover, it is important to note that a more Transformer-like architecture was employed in TRL [10], but as shown in App. B, merely translating the architecture towards Transformers without tailoring it to the specific problem did not yield significant performance improvements compared e.g. to our SPN or L2D [47]. We appreciate the reviewer's feedback and acknowledge again the ongoing need for a reference architecture for scheduling problems (see also answer A3 of reviewer 2iGC).

---

### Official Review · Reviewer_pxef · 2024-07-12

**Soundness:** 3
**Presentation:** 3
**Contribution:** 3
**Rating:** 6
**Confidence:** 4

**Summary:**

The paper proposes learning a constructive neural heuristic for the Job Shop Scheduling problem (JSP). The proposed policy network is an auto-regressive attention-based encoder-decoder model. A JSP instance is represented by a (commonly used) disjunctive graph with additional hand-crafted features. The paper proposes to train the policy network using a "self-labeling" strategy. This strategy consists in alternating for each training instance between (i) sampling a number of solutions from the current policy, selecting the one with the smallest makespan as a pseudo-label, and (ii) updating the policy using a supervised (cross entropy) loss based on the pseudo-label. The approach is tested on standard JSP benchmarks and shows superior performance to state-of-the-art neural baselines.

**Strengths:**

* The paper is clear and well written. In particular, the description of the model and the experiments is clear and detailed enough.
* In the experiments, the baselines are quite exhaustive: in addition to similar neural constructive heuristics, improvement heuristics as well as various non-learning-based approaches are considered.
* The strong performance on the Taillard and Demirkol datasets, even on instances with a number of jobs or machines not seen in training.
* The scaling of the approach is evaluated on instances with up to 100 jobs and 20 machines (versus at most 20x20 in training).

**Weaknesses:**

1. The proposed training strategy relies on stochastic sampling from the current policy to generate better-quality solutions to then improve the policy. However for a given training instance, there is no guarantee that one of the $\beta$ sampled solutions should be better than the greedy policy solution. If the solutions do not improve consistently, I can't see how the training would work.

1. Some previous works, such as [1], have shown the limitations of such random sampling in generating diverse and/or good-quality solutions, at least given a trained policy.

1. The paper has a narrow scope since the approach is tailored for the JSP. Although I agree with the authors the principles of the training could be applied to other problems, it remains to be shown if it would actually work, especially given my previous points.

1. The proposed heavy feature engineering (Tables 1 and 4) is obviously specific to the JSP and somehow goes against the end-to-end promise of neural combinatorial optimization.

[1] Chalumeau et al, Combinatorial Optimization with Policy Adaptation using Latent Space Search. NeurIPS 2023

**Questions:**

1. Did the authors monitor the quality of the generated solutions during the training? For each training instance, does the makespan of the best sampled solution consistently decrease?

1. Line 201: "we generate with the PN a set of $\beta$ different solutions." --> Is there a condition that ensures that the sampled solutions are different? In the experiments, did the author track if there are any duplicates among the $\beta$ solutions?

1. Eq (5): On the right hand-side, shouldn't it be $\bar{\pi}$ instead of $\pi$?

1. It would be interesting to discuss concurrent work [2] and previous work [3] which also propose self-improvement training strategies with related pseudo-labels.

[2] Pirnay et al, Self-Improvement for Neural Combinatorial Optimization: Sample Without Replacement, but Improvement. Transactions on Machine Learning Research (06/2024)

[3] Luo et al, Self-Improved Learning for Scalable Neural Combinatorial Optimization. arXiv:2403.19561

**Limitations:**

Yes, the limitations were addressed.

---

> ### Author Rebuttal · Authors · 2024-08-05
>
> Please find our responses to the main concerns and questions below:
> - **A1 (greedy solution concern):** We kindly disagree. If the model is highly confident in a decision, random sampling tends to align with a greedy argmax strategy. Whereas, similar to top-k and nucleus sampling (with small $k$ and $p$), enforcing (too) greedy selections can cause the model to converge prematurely to a sub-optimal policy (poor exploration issue) during training, as it only learns to amplify its sub-optimal decisions (see also Sec. 4.2.2 in your referenced [2]). Thus, **ensuring that sampled solutions always align or are better than greedy ones is not necessarily the best long-term strategy for training good models.** Empirically, if randomly sampled solutions were not better than the greedy one, we would not be able to train effective models nor produce the training curves shown in Fig. 1 of the attached PDF file (or Fig. 4 in the paper). Moreover, if that were the case, we would expect similar SPN's performance in both the greedy and randomized sections of Tab. 2 of the paper, but the results show otherwise.
> - **A2 (narrow scope):** We do agree that our model is tailored for the JSP like other models are for routing problems. However, this does not imply a narrow scope. **Effectively solving the JSP establishes a foundation for addressing other shop scheduling problems like Flow Shop (a special case of JSP) and Flexible JSP**; see also A2 of reviewer mZNb and A4 of reviewer NWh3. Furthermore, **self-labeling is a general approach that is getting applied in other CO problems**, e.g., routing ones in your referenced [2] and [3]. To back this up, we provide in Fig. 2 of the attached PDF file (above) preliminary results showing that self-labeling is indeed effective also in TSP. Thus, we are confident about the generality and wide-scope of our contribution.
> - **A3 (features):** To be precise, the features are specific to shop scheduling problems, i.e., problems having operations, jobs, and machines as entities. Note also that many of the features are taken from related published works (see e.g. App. A). **We apologize but we do not see issues in having an effective model suitable for the JSP as long as we are not claiming to have a new general end-to-end model for CO problems** (see also A3 of reviewer 2iGC). Lastly, in Section 6.3, we also proved that self-labeling allows to train well a recent end-to-end architecture (namely CL) that uses as features the processing times and machine indices only.
> - **A4 (training solution quality):** **Yes, we monitor the quality of training solutions (train.py file, line 126 of supplementary material)** to see whether the model keeps improving. Also, Fig. 4 (and the refined Fig. 1 in the attached PDF above) of our paper shows that the model improves on validation instances over the training because it improves on training instances.
> - **A5 (diverse solutions):** We agree that duplicates may happen (in very small instances), but we did check and this was not an issue. In JSP, the probability of duplicates in an instance with 10 jobs and 10 machines (smallest training instance) is in the order of 1 over $10!^{10}$, hence tiny. To further provide evidence, we use our best model (highest likelihood of producing duplicates) and count how many duplicates it generates when sampling 512 solutions (max number in our experiments). **As we count 0 duplicates in Taillard's and Demirkol's instances, we conclude that duplicates do not particularly limit our methods.**
> - **A6 (Eq. 5):** Yes, thank you for pointing this out.
> - **A7 (discuss [2] and [3]):** These recent follow-up works expand on our research in various directions. [2] uses self-labeling and proposes an advanced sampling strategy outperforming random, top-k, and nucleus sampling. The unpublished work [3] presents a local reconstruction approach to tackle large-scale routing instances, where the model is fine-tuned to reconstruct parts of a solution using self-labeling but requires pre-training with RL algorithms. **Our message is different, we prove that self-labeling can effectively train models from scratch for building complete solutions without any RL pre-training, addressing a broader and more challenging task.** Lastly, without violating anonymity, it is worth noting that these works may cite and explicitly build upon our work, which was publicly released on ArXiv before the submission. We can nevertheless include a discussion regarding these works in the paper.

---

> ### Comment · Reviewer_pxef · 2024-08-11
> **Response to authors rebuttal**
>
> I thank the authors for addressing precisely all my comments. I appreciated:
> * the additional preliminary experiments on the TSP that show the potential of the approach beyond the JSP
> * the clarifications about the random sampling, the improvements during training and the lack of duplicates for the JSP
> * the discussion of the follow-up works
>
> As the first paper which introduces self-labeling as an effective training strategy for CO, I support the acceptance of the paper.

---

### Official Review · Reviewer_2iGC · 2024-07-13

**Soundness:** 3
**Presentation:** 4
**Contribution:** 3
**Rating:** 7
**Confidence:** 4

**Summary:**

This paper introduces an effective method for learning to solve the Job Shop Scheduling Problem (JSP). The contribution is twofold: a pointer network architecture (encoder-decoder) to effectively represent the problem and an efficient learning paradigm based on self-supervised learning termed “self-labeling” in which a model is trained by supervising on the best self-generating solutions, thus not needing to collect labels. The proposed approach outperforms several SotA learning baselines.

**Strengths:**

The paper is well-written and clearly positioned in the literature. The proposed self-labeling approach, while simple, is a reasonable next step in the recent line of work of supervised approaches for combinatorial optimization, removing the reliance on optimal solutions while addressing the sparse credit assignment problem. This can be extended to other combinatorial problems. The proposed PN is not that new in terms of concept, but its execution such as feature engineering and the code implementation (which I appreciate) are pretty meaningful. The experiments are extensive in the JSP and provide clear evidence of the method’s benefits.

**Weaknesses:**

My main concern is about the applicability to other problems, which lacks experimental evidence - given this is a major point the authors make in the contributions and conclusions, I was expecting some pilot study on, say, the TSP showing the method’s applicability, but unfortunately, this was not provided. Note that given the limited time for rebuttal, I am not expecting the necessary results. Notably, there are concurrent/follow-up works that apply such an idea to other CO problems such as [1r, 2r]. Thus, I think these can make up for the lack of experiments in this area.

---

### References

[1r] Pirnay, Jonathan, and Dominik G. Grimm. "Self-Improvement for Neural Combinatorial Optimization: Sample without Replacement, but Improvement." arXiv preprint arXiv:2403.15180 (2024).

[2r] Luo, Fu, et al. "Self-Improved Learning for Scalable Neural Combinatorial Optimization." arXiv preprint arXiv:2403.19561 (2024).

**Questions:**

1. Why use random sampling instead of other techniques such as top-k and nucleus sampling? You mentioned that you made a preliminary analysis; however, results seem to be missing. According to recent literature, as [1r] above, nucleus sampling could help achieve better performance.

2. Why do you use a Pointer Network and not, for instance, re-encode step-by-step? Also, how did you choose parameters such as the number of attention heads in GAT?

**Limitations:**

Addressed in the text. Also, see the above weaknesses.

---

> ### Author Rebuttal · Authors · 2024-08-05
>
> Please find our responses to the main concerns and questions below:
> - **A1 (TSP pilot):** As noted by the reviewer, there are recent follow-up works adopting self-labeling and proving our method is applicable to other problems. We further back this claim up by including in Fig. 2 of the attached PDF file (above) the preliminary training curves comparing POMO and self-labeling on TSP. **These curves show that the renowned Attention Model for the TSP can be effectively trained with self-labeling, proving once more the applicability and generality of our method.**
>
> - **A2 (nucleus sampling):** We provide in Tab.1 below a comparison of sampling strategies at test time (due to limited time we were unable to reproduce the analysis at training). **The strategies are similar, however, top-k and nucleus sampling introduce hyperparameters that can hinder training convergence if not managed carefully (see the similar statement at the bottom of Sec. 4.2.2 in [1r]).** Consistently with policy optimization, we prefer random sampling as it has empirically shown a good balance between exploration and exploitation without introducing brittle hyperparameters. Notably, [1r] uses a more advanced sampling strategy, which is key to their performance boost. We can include in the paper a test and training time comparison of different strategies.
>
> **Tab. 1** : Comparison of random (rand), top-k, and nucleus sampling on Lawrence (LA), Taillard (TA), and Demirkol (DMU) benchmarks. For each benchmark, we report the overall average gap of the SPN when sampling $\beta = 512$ solutions. Note there are small differences but not a sampling strategy consistently better than the others.
> Sampling | LA | TA | DMU | Avg
> -----------|-----|-----|-------|------
> rand | 2.47 | 7.78 | 13.10 | 7.78
> top-3 | 2.74 | 7.67 | 13.52 | 7.97
> top-5 | 2.50 | 7.49 | 13.33 | 7.77
> nucleus (p=0.9) | 3.09 | 7.54 | 13.27 | 7.96
> nucleus (p=0.95) | 2.77 | 7.59 | 13.17 | 7.84
>
> - **A3 (why PN):** To the best of our knowledge, there is no standard reference architecture for the JSP. **We chose the well-studied PN, which has been effective in other CO problems like TSP and VRP.** However, any re-encoding strategy or generative model suitable for the problem can be used. **Note that our claim is not to propose a reference architecture for the JSP or scheduling problems, but just to have an effective one.** There is probably a need for a reference architecture for scheduling problems.
>
> - **A4 (parameters):** **We tuned hyperparameters with 5-fold cross-validation on a subset of training instances to balance performance and execution time, including the number of heads.** Although deeper (not larger) models may improve results, they also require more execution time. As constructive heuristics should be fast (metaheuristics generally run quite fast and are more powerful for CO problems, see lines 21-30), bigger models may be a drawback for their philosophy and practical applicability. Thus, we opted for a reasonable trade-off between performance and execution time.

---

> > ### Comment · Reviewer_2iGC · 2024-08-11
> > **Thanks!**
> >
> > Thanks for your reply. The authors resolved my concerns and ran additional experiments that demonstrated the applicability and validity of their approach.
> >
> > In light of this, I will raise my score and recommend the paper for acceptance.

---

### Author Rebuttal · Authors · 2024-08-05

We would like to thank all the reviewers for their valuable comments.
We are glad the **contribution** and the **significance** of our work have been recognized by all the reviewers.

Note that after the initial submission and based on your feedback, we have made the following minor modifications/inclusions to the paper:
- **We have introduced an additional benchmark encompassing small-medium instances as requested by reviewer NWh3.** This was done to complement our evaluation on instances even smaller than training ones. You can find this evaluation in Tab. 2 of the attached PDF file. Note that the SPN remains the best constructive approach even on smaller instances.
- **We have updated Fig. 4 (Sec. 6.4) of the paper with Fig. 1 in the attached PDF file.** To make the training curves less noisy with our model, we changed the batch size from 4 to 8 by keeping the same training setting described in Sec. 6.4 (i.e., that of Zhang 2020). This was done to remove the misconception that may relate the training noise to the sampling strategy (comments of reviewer pxef).

As we are currently improving and extending our work, you will also find in the attached PDF file the preliminary training curves comparing POMO and self-labeling on TSP (Fig. 2). **This figure serves to answer questions of reviewers 2iGC and pxef as well as to show that self-labeling is indeed applicable to other CO problems** (also shown in recent follow-up works).

Finally, we understand those who expressed concerns about random sampling. While it is not the optimal strategy for combinatorial optimization problems, random sampling remains a standard approach in policy optimization. Given that variations like top-k and nucleus sampling introduce brittle hyperparameters and have shown similar effectiveness (see response A2 to reviewer 2iGC), we opted for random sampling. **It is important to note that the choice of the sampling strategy is a design decision, not inherently tied to our self-labeling method.** Our rationale is that if self-labeling is effective with basic random sampling, it should also work with more advanced strategies that refine and improve upon it. Moreover, there are already recent follow-up works, such as "Self-Improvement for Neural Combinatorial Optimization: Sample Without Replacement, But Improve", that demonstrate how advanced sampling procedures can enhance training strategies for CO problems.

We hope the answers provided below effectively address the main concerns of the reviewers.

---

### Decision · Program_Chairs · 2024-09-25

**Decision:**

Accept (poster)

**Comment:**

The reviewers all support acceptance of this paper. The self-labeling strategy is novel and effective. Although other very recent works now exist using self-labeling strategies, they cannot be considered under the NeurIPS guidelines. The approach scales well and performs well against existing baselines.